# Remimazolam-Induced Anaphylaxis and Cardiovascular Collapse: A Narrative Systematic Review of Eleven Cases

**DOI:** 10.3390/medicina60060971

**Published:** 2024-06-12

**Authors:** Jaemoon Lee, Seong-Hyop Kim

**Affiliations:** 1Department of Anesthesiology and Pain Medicine, Konkuk University Medical Center, Seoul 05030, Republic of Korea; jmlee@kuh.ac.kr; 2Department of Infection and Immunology, Konkuk University School of Medicine, Seoul 05029, Republic of Korea; 3Research Institute of Medical Science, Konkuk University School of Medicine, Seoul 05029, Republic of Korea

**Keywords:** remimazolam, benzodiazepine, anaphylaxis

## Abstract

*Background and Objectives*: Remimazolam, a novel benzodiazepine, is used for procedural sedation and general anesthesia due to its rapid onset and short duration of action. However, remimazolam-induced anaphylaxis (RIA) is a rare but severe complication. This study aimed to analyze RIA characteristics, focusing on cardiovascular collapse, and provide guidelines for safe remimazolam use. *Methods*: This study conducted a systematic review using the Preferred Reporting Items for Systematic Reviews and Meta-Analyses 2020 guidelines. Research articles retrieved from PubMed on 26 May 2023, using the keywords ‘remimazolam AND anaphylaxis’ were evaluated based on the inclusion criteria of being written in English and aligning with the World Allergy Organization criteria for anaphylaxis, while studies not meeting these criteria were excluded. All published articles up to the search date were included without any date restrictions. The review analyzed factors such as age, sex, type of anesthesia, remimazolam dose (bolus/continuous), allergic symptoms and sign, epinephrine use, serum tryptase levels, and skin prick tests. *Results*: Among eleven cases, the mean age was 55.6 ± 19.6 years, with 81.8% male. Hypotension (81.8%) was the most common symptom, followed by bradycardia (54.5%) and desaturation (36.4%). Two patients experienced cardiac arrest. Serum tryptase levels confirmed anaphylaxis in ten cases. Epinephrine was the primary treatment, with intravenous doses ranging from 0.1 mg to 0.3 mg. *Conclusions*: Vigilance is crucial when administering remimazolam, adhering to recommended dosages, and promptly treating RIA with epinephrine. Further research is needed to understand the risk factors and refine the management strategies. Guidelines for safe remimazolam use are proposed.

## 1. Introduction

Perioperative anaphylaxis (PA) is rare especially during anesthesia. However, it is a serious complication that can cause a wide range of adverse effects, ranging from vascular edema and hypotension to bronchial spasm, cardiovascular collapse, and even death [1,2,3,4]. Common triggers of PA have been known to be neuromuscular blocking agents, non-steroidal anti-inflammatory drugs, and antibiotics [2,3,4,5]. The incidence of PA ranges from 1:1250 to 1:20,000 [1,3,6,7,8], with a case fatality rate ranging from 3 to 4.8% [1,2,3,5,6,9]. PA can occur at any time, but it is most observed during anesthesia, particularly during the induction phase, when a high volume of drugs of various types is administered [1,4,10].

Remimazolam is a new benzodiazepine derivative that has been developed for use in procedural sedation and general anesthesia [11,12]. It has a rapid onset and short duration of action, making it an attractive option for sedation or procedures that require general anesthesia for a short duration [13,14]. Furthermore, the use of remimazolam is steadily increasing due to the benefits of hemodynamic stability and safety with the antidote flumazenil [11,15,16]. However, as the use of remimazolam increases, the possibility of complications has emerged [17,18]. Recently, case reports of PA that occurred after the use of remimazolam have been frequently reported in South Korea, China, and Japan [17,18,19,20,21,22].

There have been reports of severe hypotension among the PAs occurring after the use of remimazolam [17,18,19,20,21,22], which contradicts the drug’s advantage of intraoperative hemodynamic stability [11,13,14,15,16]. This severe hypotension often lacks a clear pathophysiology and suggests an allergic reaction [17,18,19,20,21,22]. Therefore, there is a growing need to investigate anaphylaxis, where severe hypotension is induced by an allergic reaction following the use of remimazolam.

This study aimed to analyze the characteristics of remimazolam-induced anaphylaxis (RIA), particularly focusing on severe hypotension and cardiovascular collapse, and to provide guidelines for the safe use of remimazolam in clinical settings.

## 2. Methods

This study was approved by the Institutional Review Board of Konkuk University Medical Center (Seoul, Korea; approval no. 2023-10-002; Chairperson: Prof. Sang Heon Lee) on 6 October 2023. Research articles retrieved from PubMed on 26 May 2023, using the English keywords ‘remimazolam AND anaphylaxis’ without any filters applied were evaluated. The inclusion criteria were that (1) the entire paper had to be written in English, ensuring that all the included studies could be thoroughly evaluated and understood by the research team without language barriers and (2) the studies aligned with the criteria for anaphylaxis according to the World Allergy Organization (WAO). The WAO criteria apply when symptoms involving the skin or mucosal tissues occur along with concurrent respiratory distress, reduced blood pressure (BP), or severe gastrointestinal symptoms, or when a patient exhibits allergic symptoms after exposure to an allergen, even in the absence of typical skin involvement, and experiences hypotension, bronchospasm, or laryngeal involvement [23]. Exclusion criteria were any studies not meeting these inclusion criteria, specifically those not written in English or not aligning with the WAO anaphylaxis criteria. All published articles up to the search date were included without any date restrictions. The confirmed data were used to conduct a systematic review according to the Preferred Reporting Items for Systematic Reviews and Meta-Analyses (PRISMA) 2020 guidelines.

The following factors were taken into consideration when analyzing the cases of anaphylaxis: age, sex, country, type of anesthesia, history of allergies or immune-related diseases, infusion rate of administered remimazolam, type of surgery, presenting signs and symptoms, use of epinephrine, serum tryptase levels (acute/baseline), histopathological test, and the results of skin prick tests or intradermal tests, height, weight, total dosage of remimazolam, and administration route of epinephrine. The severity of anaphylactic reactions in each case was graded by the authors according to the 2020 WAO anaphylaxis guidelines (grade 1 = involvement of one organ system but not life-threatening; grade 2 = involvement of at least two organ systems; grade 3 = involvement of mild respiratory or gastrointestinal signs; grade 4 = involvement of severe respiratory signs; and grade 5 = circulatory or respiratory failure) [23].

The analysis estimated the cumulative number of patients with RIA and their distribution based on the selected variables. Data were reported as the mean (standard deviation) and numbers (percentages).

## 3. Results

The literature review identified a total of seven articles. Among these, six were case reports, and one was an editorial article. The six case reports, which included a total of eleven cases, were analyzed in-depth to confirm cases suitable for data analysis. All cases satisfied the anaphylaxis criteria established by the WAO and were classified as grade 5 owing to circulatory failure. A total of eleven cases were investigated in this manner (Figure 1).

Patients with RIA had a mean age of 55.6 years, with a standard deviation of 19.6 years, and 81.8% (9 out of 11) of them were male. There were three reports from South Korea, three from Japan, and one from China. Of the 11 cases, 10 occurred during general anesthesia and one during monitored anesthetic care. Two patients had a history of immune-related conditions, including one with a history of allergy to acemetacin, an antibiotic, and kikyosekko, an herbal remedy) and one with Crohn’s disease. When assessing the use of remimazolam during induction, two cases involved bolus administration, while continuous infusion was administered in five cases, and four cases used a combined method of administration (Table 1). Although not included in the table, the most common type of surgery was gastrointestinal surgery (*n* = 6), followed by orthopedic surgery (*n* = 2), thoracic surgery (*n* = 1), neck surgery (*n* = 1), and endoscopic procedures (*n* = 1).

The most frequent manifestation of RIA, in terms of the signs and symptoms, was hypotension (81.8%). Desaturation (36.4%) and bradycardia (54.5%) followed. Notably, two patients (18.2%) experienced cardiac arrest and needed advanced cardiovascular life support. A total of four patients had skin symptoms; and erythema (18.2%), rash (27.3%), and edema (18.2%) were all noted. Desaturation was the most prevalent respiratory symptom, followed by stridor, wheezing, and bronchospasm (Table 2).

All the patients received epinephrine. Only two patients received epinephrine via intramuscular (IM) injection, while all patients received epinephrine via IV administration. In four cases, epinephrine was administered continuous intravenous (IV). Serum tryptase levels in a total of ten cases and histopathological examination in one patient each provided definitive evidence of anaphylaxis. Skin prick tests were performed on nine patients, four of whom had positive results. Three patients tested positive for remimazolam. Moreover, there was no positive reaction to dextran in a skin prick test (Table 3).

## 4. Discussion

The hemodynamic stability of remimazolam is an advantage that is leading progressively to its increased use [14,24]. However, hemodynamic instability is paradoxically a prominent sign of RIA, suggesting that remimazolam may pose the greatest risk to the safety of individuals who require hemodynamic stability [25]. Detailed analyses of the causes of RIA have been limited by the small number of cases. When patients deteriorate, underreporting may occur owing to an inability to identify the cause, or it may be considered taboo to report, leading to concealment. In summary, among patients who experienced anaphylaxis, there was a trend toward significant findings in cases involving males and those who utilized the maximum recommended dose. However, confirming statistical significance was impossible because of the small number of instances. Even after performing a meta-analysis combining data from multiple studies, statistical significance could not be achieved.

Males had a higher prevalence of RIA than females, but no other specific characteristics such as age, height, weight, type of surgery, and medical history could be identified. On the other hand, women had a higher incidence of anaphylaxis during anesthesia than males [8]. In addition, a recent review identified higher BMI, older age, and low plasma albumin concentration as risk factors for delayed emergence [26]. The influence of gender is known to be minimal in the pharmacokinetic and pharmacodynamic effects of remimazolam [27]. Moreover, there is no evidence that plasma and tissue esterase, which are known to rapidly metabolize remimazolam, differ in composition or activity depending on gender. However, the clearance of remimazolam is reported to be 11% higher in females [28,29]. These variations may be due to differences in study design, patient populations, or the specific outcomes measured. Further research is needed to reconcile these findings and better understand the risk factors associated with RIA.

Hypotension was predominantly the first symptom of anaphylaxis in 81.8% of patients, a pattern like PA induced by different causes [1]. However, cutaneous and respiratory signs were less prevalent compared to cardiovascular signs. This hypotension is thought to be due to anaphylaxis-induced vasodilation rather than a cardiogenic origin. Although echocardiography was performed on only two patients, it confirmed a left ventricular hyperdynamic status. On the other hand, ST changes were observed in only two patients in the electrocardiogram conducted on all patients. Remimazolam has the advantage of hemodynamic stability and is known to be effective even in ASA PS III or IV patients [14,24]. However, if hypotension is the only feature of RIA, the prognosis could be more severe in ASA PS III or IV patients. Additionally, since accompanying signs such as cutaneous or respiratory signs may not be present, it can be difficult to determine whether hypotension is caused by RIA or the patient’s underlying conditions. This can lead to a delay in the active use of epinephrine, which is the treatment method for RIA. There is still limited knowledge about remimazolam-related hypotension. Ongoing trials aim to compare its incidence with that of propofol-induced hypotension to better understand the underlying causes [30].

Remimazolam has recommended doses of 6 mg/kg/h or 12 mg/kg/h for induction of general anesthesia and 1–2 mg/kg/h for maintenance during general anesthesia when administered as a continuous infusion [19,31,32]. For sedation, remimazolam has a dosing recommendation of 5 mg administered intravenously over a minimum of 1 min. Additionally, if a supplementary dose is needed, 2.5 mg over a minimum 15 s interval, with breaks of at least 2 min between doses, is to be administered [33,34]. Most cases of adverse reactions occur when maximum doses of remimazolam are administered rapidly. Among eleven cases, it was observed that anaphylactic reactions occurred in nine cases, when either the maximum recommended dose (12 mg/kg/h for induction) or additional bolus was used. This may be due to the lack of correlation between the remimazolam sedation scale monitoring and the occurrence of adverse reactions. The adequacy of sedation scale monitoring for remimazolam is still controversial [35,36]. Some studies suggest that sedation scale monitoring based on propofol is sufficient, while others argue that it is inadequate [25,34,35]. Furthermore, among cases of anaphylaxis with cardiac arrest or cardiovascular collapse, the sedation scale measured through an electroencephalogram appeared to be suboptimal, falling short of the general anesthesia level [18]. In many cases, despite hemodynamic instability, additional medication for amnesia and sedation was required. This could be attributed to factors that lead to the excessive use of remimazolam during induction. Additionally, RIA could be started at any moment after the induction dosage is administered, although it typically started within 10 min (data not presented). Due to the lack of sufficient cases, no statistical conclusions could be drawn.

Epinephrine was confirmed to be the first-choice drug for anaphylaxis and is generally recommended as the first-choice treatment for the management of anaphylaxis [23,37,38]. However, in cases of RIA, intravenous (IV) administration was more common than intramuscular (IM) administration. The WAO recommends initiating treatment with epinephrine at a dose of 0.3 mg to 0.5 mg for adults (0.01 mg/kg up to a maximum of 0.3 mg per does for children) via the IM route using a 1:1000 (1 mg/mL) solution as soon as anaphylaxis is diagnosed or strongly suspected [23,37,38]. Additionally, it is recommended to use a 1:10,000 (0.1 mL/mL) concentration for IV administration and to adjust the titration based on the patient’s response. However, in cases with cardiovascular collapse, IM administration may take longer to achieve the desired effect or may not be achievable. In such cases, IV administration may be considered under restricted circumstances [23,37,38]. Anaphylaxis-related fatal outcomes are associated with delayed or inadequate administration, as well as excessive administration of epinephrine, emphasizing the need for cautious and appropriate dosing [39,40]. In analysis, except for three cases, the condition improved after the use of epinephrine, and there was no need for observation in the intensive care unit after anesthesia. All eleven patients were discharged without any sequelae (data not presented in the table). Unlike other PAs, with the proactive use of epinephrine when RIA occurs, anaphylaxis-related fatalities can be avoided. Since the anesthesia environment is equipped with monitoring equipment for vital signs (electrocardiogram, peripheral saturation, non-invasive blood pressure), it is more effective to use IV administration of epinephrine rather than IM administration. This approach can help reduce sequelae after recovery from cardiovascular collapse by promptly addressing sudden changes in vital signs during RIA.

Among the 11 cases, tryptase levels were confirmed in ten cases, and elevated levels compared to baseline confirmed anaphylaxis in nine of the ten cases (positive test: acute tryptase > [(1.2 × baseline tryptase) + 2] g/L). On the other hand, only three cases were confirmed to be caused by remimazolam by a positive skin prick test, and there were no cases confirmed to be caused by dextran 40, which was mentioned as a possible cause of anaphylaxis with remimazolam [10,20]. There are a few restrictions with this, though. One drawback is that a skin pick test cannot be carried out if dextran 40 is not available, as it is in Korea. Furthermore, it has been noted that an increase in the tryptase level does not always indicate an anaphylactic reaction, because it might also indicate an anaphylactoid reaction, like that seen with dextran 40 [41,42]. However, in cases of severe hypotension after remimazolam administration, immediate discontinuation of the drug and simultaneous blood sampling to measure serum tryptase levels should be performed. This can help confirm RIA and prevent its reuse.

Sander Kempenaers et al. published a similar review [25]. Compared with the previous review, we enrolled and analyzed more patients and included not only general anesthesia but also just sedation. Additionally, unlike the previous review which covered various adverse events, this review focused solely on RIA. This allowed for a detailed analysis of its causes and provided guidelines on the use of epinephrine, the drug of choice, as well as guidelines for the use of remimazolam.

Guidelines for the Safe Use of Remimazolam

Patient AssessmentConduct a thorough medical history and allergy assessment before administration.Identify patients with a history of drug allergies, asthma, or other risk factors for anaphylaxis.Dosage and Administration: Adhere strictly to the recommended dosages.Use 6 mg/kg/h or 12 mg/kg/h for induction and 1–2 mg/kg/h for maintenance during general anesthesia with continuous infusion. For procedural sedation, administer 5 mg intravenously over a minimum of 1 min, with supplementary doses of 2.5 mg over at least 15 s with 2 min intervals.Avoid rapid administration of maximum doses to reduce the risk of adverse reactions.MonitoringContinuously monitor the cardiovascular and respiratory status, including the blood pressure, heart rate, and oxygen saturation.Utilize electrocardiography (ECG) to monitor the cardiac function, especially in patients with cardiovascular risk factors.Preparedness for AnaphylaxisC.Have emergency equipment and medications readily available, including epinephrine, antihistamines, corticosteroids, and resuscitation equipment.D.Train all medical staff in the recognition and management of anaphylaxis.Immediate ResponseE.At the first sign of anaphylaxis, immediately discontinue remimazolam and administer epinephrine intramuscularly (0.01 mg/kg, up to 0.5 mg for adults) or intravenously in cases of cardiovascular collapse.F.Follow with antihistamines and corticosteroids to manage symptoms.Post-Reaction ManagementMeasure serum tryptase levels to confirm anaphylaxis and document the reaction.Provide intensive care and continuous monitoring until the patient stabilizes.Documentation and ReportingDocument all adverse reactions thoroughly and report them to relevant health authorities to improve incidence tracking and pharmacovigilance.

## 5. Conclusions

This study underscores the importance of vigilance among anesthesiologists when administering remimazolam, particularly in recognizing and managing the risk of cardiovascular collapse. Adherence to recommended dosages and prompt administration of epinephrine in cases of RIA are crucial. Further research is essential to better understand risk factors, improve incidence tracking, and refine management strategies for RIA. Additionally, guidelines were recommended to ensure the safe use of remimazolam.

## 6. Limitations

One of the main limitations of this study is the relatively small number of cases available for analysis. While our review analyzed six reports of anaphylaxis following remimazolam use, the limited number of cases in current clinical practice at present may affect the generalizability of our findings. Although the increased use of remimazolam could potentially reveal more cases and provide a broader understanding of its adverse effects, the current data necessitate cautious interpretation.

The use of a PRISMA flow-chart, which is typically associated with systematic reviews, may not be entirely appropriate for this case series. This limitation has been acknowledged to ensure clarity and accuracy in the representation of our study. Future research with larger sample sizes and more comprehensive data will be essential to draw more robust conclusions about the risk of remimazolam-induced anaphylaxis and its management.

## Figures and Tables

**Figure 1 medicina-60-00971-f001:**
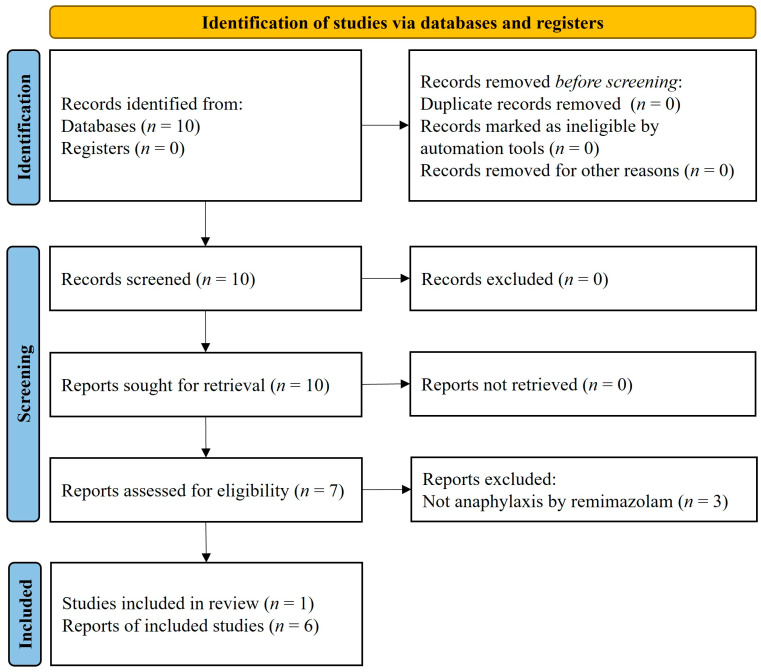
PRISMA flow diagram. Studies initially evaluated for the current pooled analysis and reasons for exclusion.

**Table 1 medicina-60-00971-t001:** Characteristics of the 11 cases of remimazolam-induced anaphylaxis during anesthesia.

Variable		Value
Age (years)		55.6 (19.6)
Female/Male		2 (18.2)/9 (81.8)
Country		
	China	1 (9.1)
	Japan	5 (45.5)
	Korea	5 (45.5)
Type of anesthesia		
	General anesthesia	10 (90.9)
	Monitored anesthetic care	1 (9.1)
Specific medical history		
	Allergic history	1 (9.1)
	Autoimmune disease	1 (9.1)
Use of remimazolam on induction		
	Bolus only	2 (18.2)
	Continuous infusion	5 (45.5)
	Combination (bolus + continuous)	4 (36.4)

Data are presented as mean (standard deviation) for age or number of patients (%) for others.

**Table 2 medicina-60-00971-t002:** Signs and symptoms of remimazolam-induced anaphylaxis.

Variable		Value
Cutaneous system		
	Erythema	2 (18.2)
	Rash	3 (27.3)
	Edema	2 (18.2)
Respiratory system		
	Desaturation	4 (36.4)
	Stridor or wheezing	2 (18.2)
	Bronchospasm	1 (9.1)
Cardiovascular system		
	Hypotension	9 (81.8)
	Tachycardia	6 (54.5)
	Cardiac arrest	2 (18.2)
	ST change	2 (18.2)

The data are presented as number of patients (%).

**Table 3 medicina-60-00971-t003:** Reported remimazolam-induced anaphylactic reactions during anesthesia.

Reference	Type	Age (Year)	Sex	Height (cm)	Weight (kg)	Medical History	Total Dose (mg)	Induction Dose	Route of Epinephrine	Acute Phase Test	Skin Test for Anaphylaxis
Hasushita [20]	GA	72	Male	166.0	61.0	Allergy (+) *	72	12 mg + 10 mg/kg/h	BIV + CIV	T (A/B)	(+): R (−): D
Hu [21]	MAC	41	Male	165.0	63.0	None	10	10 mg	BIV	H	(+): M (−): R, D
Kim [19]	GA	65	Male	177.3	75.0	None	98.8	12 mg/kg/h	CIV	T (A/B)	(−): R, M UC: D
Kim [19]	GA	69	Male	167.3	64.3	None	78	12 mg/kg/h	CIV	T (A/B)	No process
Kim [19]	GA	66	Male	165.3	53.2	None	57.4	12 mg/kg/h	BIV	T (A/B)	(−): R, M UC: D
Kim [19]	GA	23	Female	161.6	65.7	Crohn’s disease	26	12 mg/kg/h	BIV	T (A/B)	(−): R, M UC: D
Kim [19]	GA	33	Female	168.3	60.1	None	8.44	2 mg/kg/h	BIV	T (A/B)	(−): R, M UC: D
Tsurumi [17]	GA	32	Male	162.0	60.0	None	15	12 mg + 6 mg/kg/h	BIM + BIV	T (A)	(+): R, M UC: D
Uchida [18]	GA	74	Male	157.0	78.0	None	UC	4 mg + 1 mg/kg/h	BIV	T (A/B)	No process
Uchida [18]	GA	59	Male	176.0	52.0	None	9	9 mg	BIV	T (A/B)	(−): R UC: D
Yamaoka [22]	GA	78	Male	148.0	55.0	None	11	11 mg + 12 mg/kg/h	BIM + CIV	T (A/B)	(+): R UC: D

*: Acemetacin, kikyosekko. Abbreviations: GA: general anesthesia; MAC: monitored anesthetic care; UC: uncheckable; BIV: bolus in intravenous; CIV: continuous in intravenous; BIM: bolus in intramuscular; T (A): serum tryptase level in acute phase; T (A/B): serum tryptase level in acute phase and baseline; H: histopathologic test; R: remimazolam; D: dextran 40; M: midazolam.

## Data Availability

The datasets generated during and/or analyzed during the current study are available from the corresponding author on reasonable request.

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
