# Peer review of "Remimazolam-Induced Anaphylaxis and Cardiovascular Collapse: A Narrative Systematic Review of Eleven Cases"

_medicina, 2024, doi:10.3390/medicina60060971_

Round 1

Reviewer 1 Report

Comments and Suggestions for Authors

1.      The study was mixed between narrative and some features of the systematic review since the authors utilised the PRISMA flow chart. Thus, I suggest enhancing the current study to meet the criteria of a systematic review by ensuring the completion of the PRISMA checklist and completing all the needed items to consider this manuscript as a systematic review.

2.      The authors evaluate only Remimazolam-Related hypotension but not anaphylactic. Anaphylactic is multi-systemic involvement.

3.      The title is not suitable with contents since the authors highlight the CVS collapse. I suggest changing the title to Remimazolam-Related Unexpected Cardiovascular Collapse.  A systematic review.  Plz Update the manuscript accordingly

4.      Objectives are missed in the abstracts

5.      Plz mention the duration of the included studies in the abstract and the methodology.

6.      The introduction did not explain the pathophysiological mechanisms of Remimazolam-Related hypotension

7.      In the results, the first line is redundant.

8.      In line 80, plz correct 7 to seven.

9.      Figure 1. PRISMA flow diagram. PRISMA is used for systematic review.  While for narrative PRISMA you can use; Green, B.N.; Johnson, C.D.; Adams, A. Writing narrative literature reviews for peer-reviewed journals: Secrets of the trade. J. Sports Chiropr. Rehabil. 2001, 15, 5–19.

10.  In line 88, how did you calculate age, and why was the median not mean, moreover, the presentation of the median should be along with the intra-quartile range.

11.  The tables are not self-explanatory. Clearly state the meaning of each value presented in the tables, such as Mean, median, range, standard deviation, etc. for numerical data

12.  Line 129 However, “confirming statistical significance was impossible because of the small number of instances". Such numerical evidence should be explained through a metanalysis study.

13.  In line 131, the authors reported that “Males had a higher prevalence of RIA than females, but no other specific characteristics such as age, height, weight, type of surgery, and medical history could be identified”. Meanwhile, a recent review identified a higher BMI, older age and low plasma albumin concentration are risk factors for delayed emergence.   DOI: 10.1097/EJA.0000000000001902

How can you explain these variations?

14.  On page 5, third paragraph, line 140 “Hypotension was predominantly the first symptom of…”. Then in line 144, the authors stated “RIA can cause various hemodynamic disturbances, ranging from hypotension”. This statement needs reorganization.

15.  In line 150, “Remimazolam has recommended doses of 6 mg/kg/h”. For what?

16.  Line 157, “when either the maximum recommended dose or higher was used”. What is the maximum dose?

17.  In line 164, EEG, did you mean ECG? Plz define the abbreviation when appears first in the text.

18.  Line 200 “Sander Kempenaers et al. have already published a similar review (41). Compared with the previous review, we enrolled and analyzed more patients, and included not only general anesthesia but also just sedation. Moreover, we focused on treatment with epinephrine at RIA. Therefore, the present review with the previous review might provide more 203 helpful information about RIA.”

Actually, Ref 41 has evaluated all serious adverse events related to the administration of remimazolam not just CVS only, and it described pathophysiological mechanisms.

19.  The aim and the conclusion as well as the whole manuscript should be rephrased to focus on the CVS collapse side effect of remimazolam .

20.  The authors did not provide guidelines to ensure safe remimazolam use as he stated in their aims.

Comments on the Quality of English Language

Minor editing of English language required

Author Response

Reviewer #1

  1. The study was mixed between narrative and some features of the systematic review since the authors utilised the PRISMA flow chart. Thus, I suggest enhancing the current study to meet the criteria of a systematic review by ensuring the completion of the PRISMA checklist and completing all the needed items to consider this manuscript as a systematic review.

Response: Thank you for your insightful comment. We agree with your perspective and have made the necessary changes. We have revised the abstract, introduction, results, discussion, and conclusion accordingly. Additionally, we updated Figure 1 to use the PRISMA flow diagram for our analysis and a PRISMA checklist attached.

  1. The authors evaluate only Remimazolam-Related hypotension but not anaphylactic. Anaphylactic is multi-systemic involvement.

Response: Thank you for your comment. We also concur with the review's perspective that anaphylaxis is a multi-systemic involvement syndrome. Therefore, we examined other symptoms and signs of remimazolam-induced anaphylaxis, which are detailed in Table 2. Although we identified more specific symptoms and signs than the categories described in Table 2, the small number of cases prevented us from confirming statistical significance. Additionally, all cases exhibited acute hemodynamic symptoms or hypotension, which developed rapidly after allergen exposure, meeting one of the definitions of anaphylaxis. Consequently, these were classified and investigated as anaphylaxis. We will include this information in the discussion section.

→ Hypotension was predominantly the first symptom of anaphylaxis in 81.8% of patients, a pattern like PA induced by different causes [1]. However, cutaneous and respiratory signs were less prevalent compared to cardiovascular signs. This hypotension is thought to be due to anaphylaxis-induced vasodilation rather than a cardiogenic origin. Although echocardiography was performed on only two patients, it confirmed a left ventricular hyperdynamic status. On the other hand, ST changes were observed in only two patients in the electrocardiogram conducted on all patients. Remimazolam has the advantage of hemodynamic stability and is known to be effective even in ASA PS III or IV patients [14, 24]. However, if hypotension is the only feature of RIA, the prognosis could be more severe in ASA PS III or IV patients. Additionally, since accompanying signs such as cutaneous or respiratory signs may not be present, it can be difficult to determine whether hypotension is caused by RIA or the patient's underlying conditions. This can lead to a delay in the actively use of epinephrine, which is the treatment method for RIA.

  1. The title is not suitable with contents since the authors highlight the CVS collapse. I suggest changing the title to Remimazolam-Related Unexpected Cardiovascular Collapse. A systematic review. Plz Update the manuscript accordingly.

Response: Thank you for your comment. Another reviewer also mentioned this point. We have judged this work as a review and decided to conduct a systematic review according to PRISMA guidelines. However, given the small number of case reports analyzed, we have decided to include this limitation in the title as well.

→ Remimazolam-Induced Anaphylaxis and Cardiovascular Collapse: A Narrative Systematic Review on Eleven Cases

  1. Objectives are missed in the abstracts.

Response: Thank you for your comment. We will make the changes and include this information accordingly in the abstract section:

→ Background and Objectives: Remimazolam, a novel benzodiazepine, is used for procedural sedation and general anesthesia due to its rapid onset and short duration of action. However, remimazolam-induced anaphylaxis (RIA) is a rare but severe complication. This study aimed to analyze RIA characteristics, focusing on cardiovascular collapse, and provide guidelines for safe remimazolam use.

  1. Plz mention the duration of the included studies in the abstract and the methodology.

Response: Thank you for your comment. We will make the necessary changes to include the duration of the included studies in the abstract and the methodology in the abstract section and Methods section:

→ (In abstract) Methods: This study conducted a systematic review using Preferred Reporting Items for Systematic Reviews and Meta-Analyses 2020 guidelines. Research articles on 'remimazolam AND anaphylaxis' retrieved from PubMed on May 26, 2023, were assessed. Eleven cases were reviewed, considering factors like age, sex, type of anesthesia, allergies, infusion rate, type of surgery, symptoms, epinephrine use, serum tryptase levels, and skin prick tests.

→ Methods: Research articles retrieved from PubMed on May 26, 2023, using the English keywords 'remimazolam AND anaphylaxis' without any filters applied were evaluated. From these, subjects that met the following criteria were selected: (1) the entire paper was written in English, and (2) the studies aligned with the criteria for anaphylaxis according to the World Allergy Organization (WAO). The WAO criteria apply when symptoms involving the skin or mucosal tissues occur along with concurrent respiratory distress, reduced blood pressure (BP), or severe gastrointestinal symptoms, or when a patient exhibits allergic symptoms after exposure to an allergen, even in the absence of typical skin involvement, and experiences hypotension, bronchospasm, or laryngeal involvement [23]. The confirmed data were used to conduct a systematic review according to the Preferred Reporting Items for Systematic Reviews and Meta-Analyses (PRISMA) 2020 guidelines.

  1. The introduction did not explain the pathophysiological mechanisms of Remimazolam-Related hypotension.

Response: Thank you for your comment. The pathophysiological mechanism of remimazolam-induced hypotension remains unknown. However, it is considered one of the signs of an allergic reaction. This was mentioned in the introduction section:

→ There have been reports of severe hypotension among the PAs occurring after the use of remimazolam [17-22], which contradicts the drug's advantage of intraoperative hemodynamic stability [11, 13-16]. This severe hypotension often lacks a clear pathophysiology and suggests an allergic reaction [17-22]. Therefore, there is a growing need to investigate anaphylaxis, where severe hypotension is induced by an allergic reaction following the use of remimazolam.

  1. In the results, the first line is redundant.

Response: Thank you for your comment. We will remove the redundant content.

→ The literature review identified a total of seven articles.

  1. In line 80, plz correct 7 to seven.

Response: Thank you for your comment. We will change the content.

→ The literature review identified a total of seven articles. Among these, six were case reports, and one was an editorial article. The six case reports were analyzed in-depth to confirm cases suitable for data analysis. All cases satisfied the anaphylaxis criteria established by the WAO were classified as grade 5 owing to circulatory failure. A total of eleven cases were investigated in this manner (Figure 1).

  1. Figure 1. PRISMA flow diagram. PRISMA is used for systematic review. While for narrative PRISMA you can use; Green, B.N.; Johnson, C.D.; Adams, A. Writing narrative literature reviews for peer-reviewed journals: Secrets of the trade. J. Sports Chiropr. Rehabil. 2001, 15, 5–19.

Response: Thank you for your comment. We will change the figure 1.

  1. In line 88, how did you calculate age, and why was the median not mean, moreover, the presentation of the median should be along with the intra-quartile range.

Response: Thank you for your comment. The total number of patients is eleven, and each data was confirmed in the case report. The ages of the patients were as followings: 23, 32, 33, 41, 59, 65, 66, 69, 72, 74, and 78 years. After performing the Kolmogorov-Smirnov test and the Shapiro-Wilk test, normality was confirmed, so the measures were changed from median to mean. Additionally, the interquartile range was replaced with standard deviation.

→ Patients with RIA had a mean age of 55.6 years, with a standard deviation of 19.6 years, and 81.8% (9 out of 11) of them were male.

Table 1.

Variable

Value

Age [years, IQR]

55.6 [19.6]

~

~

Data are presented as number (%), mean [standard deviation]

  1. The tables are not self-explanatory. Clearly state the meaning of each value presented in the tables, such as Mean, median, range, standard deviation, etc. for numerical data.

Response: Thank you for your comment. We will make changes to reflect this in table 1, table 2:

Table 1.

Variable

Value

Age [years, IQR]

Median: 65 [37-70.5]

~

~

Data are presented as number (%), median [interquartile range]

Table 2. Signs and symptoms of remimazolam-induced anaphylaxis

Variable

Value

Cutaneous system

  1. Line 129 However, “confirming statistical significance was impossible because of the small number of instances". Such numerical evidence should be explained through a metanalysis study.

Response: Thank you for your comment. The number of patients in each case report was small, and even when combining the data from the case series, statistical significance could not be confirmed. We will make changes to reflect this in discussion section:

→ However, confirming statistical significance was impossible due to the small number of instances. Even after performing a meta-analysis combining data from multiple studies, statistical significance could not be achieved.

  1. In line 131, the authors reported that “Males had a higher prevalence of RIA than females, but no other specific characteristics such as age, height, weight, type of surgery, and medical history could be identified”. Meanwhile, a recent review identified a higher BMI, older age and low plasma albumin concentration are risk factors for delayed emergence. DOI: 10.1097/EJA.0000000000001902

How can you explain these variations?

Response: Thank you for your comment. Our findings show statistically notable trends; however, due to the insufficient number of cases, further research is necessary. In contrast, the systematic review mentioned by the reviewer includes 68 patients with delayed emergence, highlighting a difference in case numbers. Additionally, other articles in this journal discussing anaphylaxis did not identify any significant statistical significance either. We will add this information and revise the discussion accordingly.

→ Males had a higher prevalence of RIA than females, but no other specific characteristics such as age, height, weight, type of surgery, and medical history could be identified. On the other hand, women had a higher incidence of anaphylaxis during anesthesia than males [8]. And a recent review identified higher BMI, older age, and low plasma albumin concentration as risk factors for delayed emergence [25]. The influence of gender is known to be minimal in the pharmacokinetic and pharmacodynamic effects of remimazolam [26]. Moreover, there is no evidence that plasma and tissue esterase, which are known to rapidly metabolize remimazolam, differ in composition or activity depending on gender. However, since clearance of remimazolam is reported to be 11% higher in females [27, 28]. These variations may be due to differences in study design, patient populations, or the specific outcomes measured. Further research is needed to reconcile these findings and better understand the risk factors associated with RIA.

  1. On page 5, third paragraph, line 140 “Hypotension was predominantly the first symptom of…”. Then in line 144, the authors stated “RIA can cause various hemodynamic disturbances, ranging from hypotension”. This statement needs reorganization.

Response: Thank you for your comment. This text aims to emphasize that remimazolam-induced anaphylaxis is characterized by prominent cardiovascular signs compared to other perioperative anaphylaxis. Even in the absence of cutaneous or respiratory signs, persistent hypotension alone should raise suspicion. Remimazolam is known for its hemodynamic stability, even in patients with ASA PS III or higher. However, in such patients, the occurrence of RIA can be more critical, and it may be challenging to distinguish whether hypotension is caused by the patient's inherent condition or by RIA-induced hemodynamic instability. In this case, instead of searching for other signs of anaphylaxis, prompt administration of epinephrine and correction of hemodynamic instability should be prioritized in discussion section:

→ This hypotension is thought to be due to anaphylaxis-induced vasodilation rather than a cardiogenic origin. Although echocardiography was performed on only two patients, it confirmed a left ventricular hyperdynamic status. On the other hand, ST changes were observed in only two patients in the electrocardiogram conducted on all patients. Remimazolam has the advantage of hemodynamic stability and is known to be effective even in ASA PS III or IV patients [14, 24]. However, if hypotension is the only feature of RIA, the prognosis could be more severe in ASA PS III or IV patients. Additionally, since accompanying signs such as cutaneous or respiratory signs may not be present, it can be difficult to determine whether hypotension is caused by RIA or the patient's underlying conditions. This can lead to a delay in the actively use of epinephrine, which is the treatment method for RIA.

  1. In line 150, “Remimazolam has recommended doses of 6 mg/kg/h”. For what?

Response: Thank you for your comment. We will adjust the sentences to clearly indicate that each dosage is used for the induction and maintenance of general anesthesia in discussion section:

→ Remimazolam has recommended doses of 6 mg/kg/h or 12 mg/kg/h for induction of general anesthesia and 1-2 mg/kg/h for maintenance during general anesthesia when administered as a continuous infusion [19, 29, 30].

  1. Line 157, “when either the maximum recommended dose or higher was used”. What is the maximum dose?

Response: Thank you for your comment. We will revise the sentences to address the unclear maximum recommended dosage in discussion:

→ Most cases of adverse reactions occur when maximum doses of remimazolam are administered rapidly. Among eleven cases, it was observed that anaphylactic reactions occurred in nine cases, when either the maximum recommended dose (12 mg/kg/h for induction) or additional bolus was used.

  1. In line 164, EEG, did you mean ECG? Plz define the abbreviation when appears first in the text.

Response: Thank you for your comment. The EEG mentioned here refers to proceed electroencephalogram. We will revise the content accordingly in discussion section:

→ Furthermore, among cases of anaphylaxis with cardiac arrest or cardiovascular collapse, the sedation scale measured through procced electroencephalogram appeared to be suboptimal, falling short of the general anesthesia level.

  1. Line 200 “Sander Kempenaers et al. have already published a similar review (41). Compared with the previous review, we enrolled and analyzed more patients, and included not only general anesthesia but also just sedation. Moreover, we focused on treatment with epinephrine at RIA. Therefore, the present review with the previous review might provide more helpful information about RIA.” Actually, Ref 41 has evaluated all serious adverse events related to the administration of remimazolam not just CVS only, and it described pathophysiological mechanisms.

Response: Thank you for your comment. The reviewer mentioned that the referenced paper correctly identifies serious adverse events induced by remimazolam. However, the paper does not adequately address the pathophysiological mechanisms involved. Additionally, as the paper covers various adverse events, it lacks detailed information on each specific event. Furthermore, the paper does not provide guidelines or appropriate treatment methods. In this paper, we aim to focus solely on remimazolam-induced anaphylaxis, providing guidelines and treatment methods. We will revise the sentences to make this distinction clearer in discussion section:

→ Sander Kempenaers et al. have already published a similar review [25]. Compared with the previous review, we enrolled and analyzed more patients, and included not only general anesthesia but also just sedation. Additionally, unlike the previous review which covered various adverse events, this review focuses solely on RIA. This allows for a detailed analysis of its causes and provides guidelines on the use of epinephrine, the drug of choice, as well as guidelines for the use of remimazolam.

  1. The aim and the conclusion as well as the whole manuscript should be rephrased to focus on the CVS collapse side effect of remimazolam.

Response: Thank you for your comment. We have revised to focus on cardiovascular collapse in the discussion and conclusion section:

→ This study underscores the importance of vigilance among anesthesiologists when administering remimazolam, particularly in recognizing and managing the risk of cardiovascular collapse. Adherence to recommended dosages and prompt administration of epinephrine in cases of RIA are crucial. Further research is essential to better understand risk factors, improve incidence tracking, and refine management strategies for RIA. Additionally, the following guidelines are recommended to ensure safe use of remimazolam:

  1. The authors did not provide guidelines to ensure safe remimazolam use as he stated in their aims.

Response: Thank you for your comment. We will add the guideline and change the aim accordingly in introduction section and conclusion section:

→ Introduction: This study aimed to analyze the characteristics of remimazolam-induced anaphylaxis (RIA), particularly focusing on cardiovascular collapse, and to provide guidelines to ensure the safe use of remimazolam.

Guidelines for Safe Use of Remimazolam

1) Patient Assessment:

Conduct a thorough medical history and allergy assessment before administration.

Identify patients with a history of drug allergies, asthma, or other risk factors for anaphylaxis.

2) Dosage and Administration:

Adhere strictly to the recommended dosages: 6 mg/kg/h or 12 mg/kg/h for induction and 1-2 mg/kg/h for maintenance during general anesthesia with continuous infusion. For procedural sedation, administer 5 mg intravenously over a minimum of 1 minute, with supplementary doses of 2.5 mg over at least 15 seconds with 2-minute intervals.

Avoid rapid administration of maximum doses to reduce the risk of adverse reactions.

3) Monitoring:

Continuously monitor cardiovascular and respiratory status, including blood pressure, heart rate, and oxygen saturation.

Utilize electrocardiography (ECG) to monitor cardiac function, especially in patients with cardiovascular risk factors.

4) Preparedness for Anaphylaxis:

Have emergency equipment and medications readily available, including epinephrine, antihistamines, corticosteroids, and resuscitation equipment.

Train all medical staff in the recognition and management of anaphylaxis.

5) Immediate Response:

              At the first sign of anaphylaxis, immediately discontinue remimazolam and administer epinephrine intramuscularly (0.01 mg/kg, up to 0.5 mg for adults, 0.3 mg for children) or intravenously in cases of cardiovascular collapse.

Follow with antihistamines and corticosteroids to manage symptoms.

6) Post-Reaction Management:

              Measure serum tryptase levels to confirm anaphylaxis and document the reaction.

Provide intensive care and continuous monitoring until the patient stabilizes.

7) Documentation and Reporting:

              Document all adverse reactions thoroughly and report them to relevant health authorities to improve incidence tracking and pharmacovigilance.

Reviewer 2 Report

Comments and Suggestions for Authors

Dear authors,
your review article is written very professionally and brings very useful knowledge about the potential risk of anaphylaxis when using remimazolam. I suggest that you comment in the discussion on which doses of epinephrine were administered intravenously in the treatment of an allergic reaction to the drug and correlate them with the mentioned WAO recommendations for intramuscular administration.

Author Response

Reviewer #2

  1. your review article is written very professionally and brings very useful knowledge about the potential risk of anaphylaxis when using remimazolam. I suggest that you comment in the discussion on which doses of epinephrine were administered intravenously in the treatment of an allergic reaction to the drug and correlate them with the mentioned WAO recommendations for intramuscular administration.

Response: Thank you for your insightful comment. In the case reports, we reviewed how epinephrine was administered to each patient, but the exact dosages were not always fully detailed. However, it was noted that in most cases, the administration did not stop with intramuscular (IM) use but extended to intravenous (IV) use, and hemodynamic instability was corrected after the use of continuous IV infusion. The method of epinephrine administration recommended by the WAO is IM injection. The dosage for adults is 0.3 to 0.5 mg, which can be repeated every 5 to 15 minutes. IV administration is also possible, and it is recommended to titrate based on the patient's response.

In the case of remimazolam, which is used for general anesthesia or sedation, the drug is typically administered with monitoring equipment (electrocardiogram, blood pressure monitoring, pulse oximetry) in place. We believe that in the event of RIA, the use of continuous IV infusion of epinephrine could be considered before IM use. This aspect has been addressed in the discussion section and the guidelines.

→ The WAO recommends initiating treatment with epinephrine at a dose of 0.3mg to 0.5 mg for adults (0.01mg/kg up to a maximum of 0.3 mg per does for children) via the IM route using a 1:1000 (1 mg/mL) solution as soon as anaphylaxis is diagnosed or strongly suspected [23, 36, 37]. Additionally, it is recommended to use a 1:10000 (0.1 ml/mL) concentration for IV administration and to adjust the titration based on the patient's response. However, in cases with cardiovascular collapse, IM administration may take longer to achieve the desired effect or may not be achievable. In such cases, IV administration may be considered under restricted circumstances [23, 36, 37]. Anaphylaxis-related fatal outcomes are associated with delayed or inadequate administration, as well as excessive administration of epinephrine, emphasizing the need for cautious and appropriate dosing [38, 39]. In analysis, except for three cases, the condition improved after the use of epinephrine, and there was no need for observation in the intensive care unit after anesthesia. All eleven patients were discharged without any sequelae (data not presented in the table). Unlike other PAs, the proactive use of epinephrine when RIA occurs, anaphylaxis-related fatalities can be avoided. Since the anesthesia environment is equipped with monitoring equipment for vital signs (electrocardiogram, peripheral saturation, non-invasive blood pressure), it is more effective to use IV administration of epinephrine rather than IM administration. This approach can help reduce sequelae after recovery from cardiovascular collapse by promptly addressing sudden changes in vital signs during RIA.

Reviewer 3 Report

Comments and Suggestions for Authors

The authors analysis 6 report of anaphylaxis after remimazolam use.

The main limitation of the study is the few number of case at the moment in clinical practice. Maybe the spread in the use of remimazolam can increase this effects but at the moment I will be very carful.

I suggest to the authors to report in the title a case series and not to use the prisma flow-chart

Best Regards

Author Response

Reviewer #3

  1. The authors analysis 6 report of anaphylaxis after remimazolam use. The main limitation of the study is the few number of case at the moment in clinical practice. Maybe the spread in the use of remimazolam can increase this effects but at the moment I will be very carful. I suggest to the authors to report in the title a case series and not to use the prisma flow-chart.

Response: Thank you for your insightful comment. Reviewer #1 and Reviewer #2 suggested using PRISMA to write a systematic review or narrative review rather than a case series. However, we also agree with your opinion. We will this in the limitations section:

→ Limitation

One of the main limitations of this study is the relatively small number of cases available for analysis. While our review analyzed six reports of anaphylaxis following remimazolam use, the limited number of cases in clinical practice at present may affect the generalizability of our findings. Although the spread and increased use of remimazolam could potentially reveal more cases and provide a broader understanding of its adverse effects, the current data necessitate cautious interpretation.

The use of a PRISMA flow-chart, which is typically associated with systematic reviews, may not be entirely appropriate for this case series. This limitation has been acknowledged to ensure clarity and accuracy in the representation of our study. Future research with larger sample sizes and more comprehensive data will be essential to draw more robust conclusions about the risk of remimazolam-induced anaphylaxis and its management.

Round 2

Reviewer 1 Report

Comments and Suggestions for Authors

Thank you for your response to my comments on your manuscript. I appreciate your efforts in addressing some of the issues I raised. However, I regret to inform you that the manuscript still requires significant work before it can be considered for publication.

There are still several critical concerns that have not been adequately addressed. Specifically:

1-As a systematic review, there should be inclusion and exclusion criteria. 

2- The author did not answer my comment #5. "Plz mention the duration of the included studies in the abstract and the methodology". It means the duration for screening of the analyzed studies and is not your duration. If you did not allocate duration plz indicate that these are all the published articles.

3- The author did not answer my comment #6. The introduction did not explain the pathophysiological mechanisms of Remimazolam-Related hypotension.

I am referring you to 

https://journals.plos.org/plosone/article?id=10.1371/journal.pone.0275451

Moreover, you write some mechanisms in your response #14

4-Page 2 line 47 is redundant. Try to close up the introduction with the gap of knowledge and study aims, not to reporting the results

5-Figure 1 and Prisam's record  showed seven cases while the title as well as in the manuscript (lines 92-93 there are 11 cases, and Tables 1 and 3) are controversial:

The author did not utilize the PRISMA efficiently.

6-The researcher did not answer comment 11. The tables are not self-explanatory. Clearly state the meaning of each value presented in the tables, such as Mean, median, range, standard deviation, etc. for numerical data.

The values in Table 1 are still not clear, I do not know which one is number (%), and which is for mean [standard deviation]. I would suggest defining the value with variables such as what you did for age. 

In the same Table Age [years, IQR]. I think you mean Age [meadin, IQR], if so, the table's legend should be updated. Moreover, the author's response in this regard differs from what has been mentioned in the updated version.

7-Guidelines come before the conclusion.

 8-Line 277 page 8 "The use of a PRISMA flow-chart, which is typically associated with systematic reviews, may not be entirely appropriate for this case series" Why and based on what?

9-I will be satisfied if the authors address all the comments and attach with his response a PRISMA Checklist 

 https://www.prisma-statement.org/prisma-2020-checklist

to fulfil the requirements of the systematic review.

Comments on the Quality of English Language

Minor corrections

Author Response

  1. As a systematic review, there should be inclusion and exclusion criteria.

Response: Thank you for your comment. We have revised it to include the inclusion criteria and exclusion criteria in the methods section and the abstract:

→ Methods (abstract): This study conducted a systematic review using Preferred Reporting Items for Systematic Reviews and Meta-Analyses 2020 guidelines. Research articles retrieved from Pubmed on May 26, 2023, using the keywords 'remimazolam AND anaphylaxis' were evaluated based on the inclusion criteria of being written in English and aligning with the World Allergy Organization criteria for anaphylaxis, while studies not meeting these criteria were excluded. All published articles up to the search date were included without any date restrictions. The review analyzed factors such as age, sex, type of anesthesia, allergies, infusion rate, type of surgery, symptoms, epinephrine use, serum tryptase levels, and skin prick tests.

→ Methods: Research articles retrieved from PubMed on May 26, 2023, using the English keywords 'remimazolam AND anaphylaxis' without any filters applied were evaluated. Inclusion criteria were: (1) the entire paper had to be written in English, ensuring that all included studies could be thoroughly evaluated and understood by the research team without language barriers, and (2) the studies aligned with the criteria for anaphylaxis according to the World Allergy Organization (WAO). The WAO criteria apply when symptoms involving the skin or mucosal tissues occur along with concurrent respiratory distress, reduced blood pressure (BP), or severe gastrointestinal symptoms, or when a patient exhibits allergic symptoms after exposure to an allergen, even in the absence of typical skin involvement, and experiences hypotension, bronchospasm, or laryngeal involvement [23]. Exclusion criteria were any studies not meeting these inclusion criteria, specifically those not written in English or not aligning with the WAO anaphylaxis criteria. All published articles up to the search date were included without any date restrictions. The confirmed data were used to conduct a systematic review according to the Preferred Reporting Items for Systematic Reviews and Meta-Analyses (PRISMA) 2020 guidelines.

  1. The author did not answer my comment #5. "Plz mention the duration of the included studies in the abstract and the methodology". It means the duration for screening of the analyzed studies and is not your duration. If you did not allocate duration plz indicate that these are all the published articles.

Response: Thank you for your comment. We have revised it to include the fact that all papers published up to the search date were included, with no restrictions on the publication year in the methods section and the abstract:

→ Methods (abstract): This study conducted a systematic review using Preferred Reporting Items for Systematic Reviews and Meta-Analyses 2020 guidelines. Research articles retrieved from Pubmed on May 26, 2023, using the keywords 'remimazolam AND anaphylaxis' were evaluated based on the inclusion criteria of being written in English and aligning with the World Allergy Organization criteria for anaphylaxis, while studies not meeting these criteria were excluded. All published articles up to the search date were included without any date restrictions. The review analyzed factors such as age, sex, type of anesthesia, allergies, infusion rate, type of surgery, symptoms, epinephrine use, serum tryptase levels, and skin prick tests.

→ Methods: Research articles retrieved from PubMed on May 26, 2023, using the English keywords 'remimazolam AND anaphylaxis' without any filters applied were evaluated. Inclusion criteria were: (1) the entire paper had to be written in English, ensuring that all included studies could be thoroughly evaluated and understood by the research team without language barriers, and (2) the studies aligned with the criteria for anaphylaxis according to the World Allergy Organization (WAO). The WAO criteria apply when symptoms involving the skin or mucosal tissues occur along with concurrent respiratory distress, reduced blood pressure (BP), or severe gastrointestinal symptoms, or when a patient exhibits allergic symptoms after exposure to an allergen, even in the absence of typical skin involvement, and experiences hypotension, bronchospasm, or laryngeal involvement [23]. Exclusion criteria were any studies not meeting these inclusion criteria, specifically those not written in English or not aligning with the WAO anaphylaxis criteria. All published articles up to the search date were included without any date restrictions. The confirmed data were used to conduct a systematic review according to the Preferred Reporting Items for Systematic Reviews and Meta-Analyses (PRISMA) 2020 guidelines.

  1. The author did not answer my comment #6. The introduction did not explain the pathophysiological mechanisms of Remimazolam-Related hypotension. I am referring you to https://journals.plos.org/plosone/article?id=10.1371/journal.pone.0275451 Moreover, you write some mechanisms in your response #14.

Response: The pathophysiology of remimazolam-related hypotension has not yet been clearly elucidated. Additionally, this article is not limited to remimazolam-related hypotension but focused on cases of cardiovascular collapse. So, we felt it was not suitable to describe this in the introduction. However, remimazolam-related hypotension is an adverse effect that can frequently occur before cardiovascular collapse, which was discussed in the discussion section. Furthermore, the mentioned trial was also addressed in the discussion.

→ This can lead to a delay in the actively use of epinephrine, which is the treatment method for RIA. There is still limited knowledge about remimazolam-related hypotension. Ongoing trials aim to compare its incidence with that of propofol-induced hypotension to better understand the underlying causes[Yokose, 2022 #150].

  1. Page 2 line 47 is redundant. Try to close up the introduction with the gap of knowledge and study aims, not to reporting the results.

Response: We have revised the study aim. However, I did not fully understand your exact intention. If further revisions are needed, please provide more detailed information on the changes required.

→ This study aimed to analyze the characteristics of remimazolam-induced anaphylaxis (RIA), particularly focusing on severe hypotension and cardiovascular collapse, and to provide guidelines for the safe use of remimazolam in clinical settings.

  1. Figure 1 and Prisam's record showed seven cases while the title as well as in the manuscript (lines 92-93 there are 11 cases, and Tables 1 and 3) are controversial.

Response: Thank you for your comment. A total of 6 case reports were analyzed, and since some case reports included multiple cases, a total of 11 cases were ultimately analyzed. We have revised it in result section, and we have changed the figure 1:

→ The six case reports, which included a total of eleven cases, were analyzed in-depth to confirm cases suitable for data analysis.

  1. The researcher did not answer comment 11. The tables are not self-explanatory. Clearly state the meaning of each value presented in the tables, such as Mean, median, range, standard deviation, etc. for numerical data.

Response: Thank you for your comment. We will revise it to include this in table 1, table 2:

Table 1. Characteristics of the 11 cases of remimazolam-induced anaphylaxis during anesthesia

Variable

Value

Age (years)

55.6 (19.6)

Female / Male

2 (18.2) / 9 (81.8)

Data are presented as mean (standard deviation) for age or number of patients (%) for others.

Table 2. Signs and symptoms of remimazolam-induced anaphylaxis

Variable

Value

Cutaneous system

The data are presented as number of patients (%).

  1. Guidelines come before the conclusion.

Response: Thank you for your comment. We will revise it.

  1. Line 277 page 8 "The use of a PRISMA flow-chart, which is typically associated with systematic reviews, may not be entirely appropriate for this case series" Why and based on what?

Response: Thank you for your comment. This content is based on the feedback of another reviewer, who suggested that the limitation should note that the article analyzed eleven cases, making it more suitable for a case series rather than a systematic review using PRISMA.

  1. I will be satisfied if the authors address all the comments and attach with his response a PRISMA Checklist

Response: Thank you for your comment. WE will draft and include this content.